# CONTEXTUALIZED ROLE INTERACTION FOR NEURAL MACHINE TRANSLATION

## ABSTRACT

Word inputs tend to be represented as single continuous vectors in deep neural networks. It is left to the subsequent layers of the network to extract relevant aspects of a word's meaning based on the context in which it appears. In this paper, we investigate whether word representations can be improved by explicitly incorporating the idea of latent roles. That is, we propose a role interaction layer (RIL) that consists of context-dependent (latent) role assignments and role-specific transformations. We evaluate the RIL on machine translation using two language pairs (En-De and En-Fi) and three datasets of varying size. We find that the proposed mechanism improves translation quality over strong baselines with limited amounts of data, but that the improvement diminishes as the size of data grows, indicating that powerful neural MT systems are capable of implicitly modeling role-word interaction by themselves. Our qualitative analysis reveals that the RIL extracts meaningful context-dependent roles and that it allows us to inspect more deeply the internal mechanisms of state-of-the-art neural machine translation systems.

## 1 INTRODUCTION

Existing deep learning approaches to natural language processing and machine translation (MT) are usually constructed as a two-stage process. In the first stage, each discrete token in an input sentence is converted into a continuous vector via a table lookup, resulting in a sequence of token representations. This sequence is then processed by the main part of the network for solving the problem at hand. In this paper, we focus on the first stage, the token representation.

The standard table lookup, or word embedding layer, embeds each token into a high-dimensional, continuous vector space independently from the other tokens in the input. It was noticed earlier, for instance by Bengio et al. (2003) and Choi et al. (2017), that this high-dimensional token vector encodes multiple aspects of the token's meaning. It is then left for the subsequent layers of a neural network to accurately extract one of these aspects based on the context. This approach has been successful in applications such as machine translation (Bahdanau et al., 2015; Gehring et al., 2017; Vaswani et al., 2017), where a lot of data is available and powerful architectures, such as recurrent networks, convolutional networks and self-attention, are used for the subsequent layers.

In this paper, we examine whether we can enhance token representations by augmenting networks with a novel layer that captures and resolves the latent roles of each token based on the context. This layer, which we call a "role interaction layer" (RIL), works on top of any sequence of vectors, by first extracting the role assignment of each token based on the entire context and then transforming each token vector representation based on its assigned role(s). More specifically, we introduce a small recurrent neural network that succinctly represents the context and subsequently determines the role assignment of each token using one of three strategies: dense, additive and sparse assignment. Each role has a trainable transformation matrix that rotates and scales the corresponding token's vector representation. These transformed role-specific vectors are then combined by a weighted sum based on the role assignment and fed into the subsequent layers.

We extensively evaluate the proposed RIL on machine translation using two language pairs (En-De and En-Fi) and three datasets with varying sizes (IWSLT'14 De-En–160k sentence pairs, WMT'14 En-De–4.5M sentence pairs, and WMT'17 En-Fi–2.3M sentence pairs). We test variants of the proposed approach against the baselines trained with the same setup. We observe that the proposed

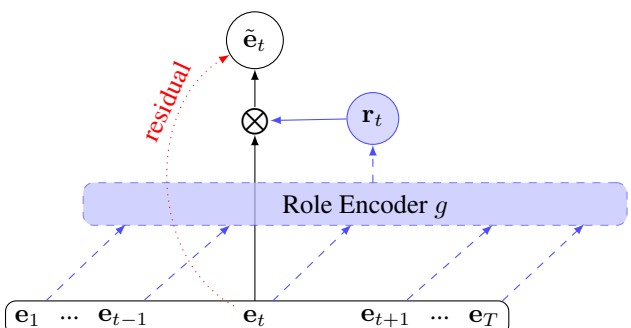

Figure 1: Illustration of proposed role interaction layer. A role encoder function $g$ (in our work, a CNN or LSTM) computes a contextual role representation $\mathbf{r}_t$ which is used to modulate representation $\mathbf{e}_t$ resulting in $\tilde{\mathbf{e}}_t$. The additive, residual connection (dotted line) is optional.

RIL, which explicitly models latent roles, indeed improves translation quality over a strong baseline when there is only limited amount of training data. The improvement, however, diminishes as the amount of data grows. On the large scale WMT'14 En-De dataset, improvements are small compared to our carefully tuned baseline, indicating that role interaction can indeed be captured implicitly by powerful models with enough training data. However, qualitative analysis reveals that the proposed RIL does still extract meaningful roles that are highly context-dependent, giving us an insight into the internal mechanism of state-of-the-art neural machine translation systems.

## 2   BACKGROUND: NEURAL MACHINE TRANSLATION

Most existing neural machine translation systems, i.e., recurrent (Bahdanau et al., 2015), convolutional (Gehring et al., 2017) and self-attention-based systems (Vaswani et al., 2017), are special cases of a conditional neural language models that consist of an encoder and a decoder. The encoder takes as input a sequence of source-side tokens $X = (x_1, \ldots, x_T)$, and first turns it into a sequence of token vectors:

$$\mathbf{E}_X = (\mathbf{e}_1^X, \ldots, \mathbf{e}_T^X), \tag{1}$$

where $\mathbf{e}_t^X = f_{\text{emb}}^X(x_t)$. It then processes this sequence into a set of vector representations $H$ to be used by the decoder. The decoder on the other hand takes as input a partial prefix of the target side tokens at each time step along with the encoded representation $H$: $Y_t = (y_1, \ldots, y_{t-1})$. Similarly to the encoder, the prefix $Y_t$ is first transformed into a sequence of (target) token vectors:

$$\mathbf{E}_Y = (\mathbf{e}_1^Y, \ldots, \mathbf{e}_{t-1}^Y), \tag{2}$$

where $\mathbf{e}_t^Y = f_{\text{emb}}^Y(y_t)$, similarly to the encoder. Given $Y_t$ and $H$, the decoder computes the distribution over all possible target tokens, $\log p(y_t | Y_t, H)$.

By summing these log-probabilities over the target sentence, we get the log-probability over the entier sentence $Y$ given a source sentence $X$:

$$\log p(Y|X) = \sum_{t=1}^{T'} \log p(y_t | y_{<t}, X). \tag{3}$$

We train this neural machine translation system by maximizing the log-likelihood over a training set consisting of source-target pairs: $\max \sum_{n=1}^{n} \sum_{t=1}^{T_n'} \log p(y_t^n | y_{<t}^n, X^n)$.

## 3   MODULATING TOKEN REPRESENTATION WITH CONTEXTUAL ROLES

We focus on token embedding in Eqs. (1)–(2). A usual approach to embedding a discrete token into a continuous space is to use a so-called table lookup or embedding layer (Bengio et al., 2003), which is equivalent to multiplying an embedding matrix with a one-hot vector of a token:

$$f_{\text{emb}}^X(x_t) = W_{\text{emb}}^X \mathbf{1}_{x_t},$$

where $W_{\text{emb}}^X \in \mathbb{R}^{h \times |V_X|}$ is an embedding matrix and $\mathbf{1}_{x_t}$ is a one-hot vector which is has value 1 only in dimension $x_t$ and is 0 otherwise. This effectively assigns one trainable $h$-dimensional continuous vector to each token, regardless of the context in which the token appears.[1]

---

[1] The same applies to the target side by replacing $X$ with $Y$.

This high-dimensional, distributed representation encodes multiple aspects of the token's underlying meaning. Choi et al. (2017) demonstrated this property by applying PCA to the continuous vectors of a set of neighbouring tokens given a target token and inspecting the neighbouring tokens lying along each principal axis. In their analysis, for instance, along the first principal axis of "notebook" lied "notebooks", "sketchbooks", "diary" and "jottings", while "laptop", "ipaq" and "palmtop" lied along the second principal axis. In other words, various aspects/meanings of a single token are entangled into a single continuous vector in this conventional approach to token embedding, and it is up to follow-up layers in the network to disentangle and select a relevant subset of these aspects.

Instead, we propose to augment this embedding layer with a specialized mechanism for separately modelling different aspects of tokens based on the context. We first define a role assignment of each token as a low-dimensional vector $\mathbf{r}_t \in \mathbb{R}^{h_r}$, where $h_r \ll h$.[2] The role assignment vector is computed based on the context, in our case a small recurrent neural network with long short-term memory (LSTM) units (Hochreiter & Schmidhuber, 1997) followed by a single nonlinear layer:

$$(\mathbf{r}_1, \mathbf{r}_2, \ldots, \mathbf{r}_T) = f_{\text{role}}(\text{LSTM}(\mathbf{e}_1, \mathbf{e}_2, \ldots, \mathbf{e}_T)). \tag{4}$$

Each dimension $i \in \{1, \ldots, h_r\}$ of the role assignment vector corresponds to the degree to which the corresponding token is assigned to the $i$-th role. Each role $i$ is associated with a distinct trainable transformation matrix $U_i \in \mathbb{R}^{h \times h}$. This transformation matrix rotates and scales the token representation $\mathbf{e}_t$ in its own way, and the transformed token representations are combined by

$$\tilde{\mathbf{e}}_t = \sum_{i=1}^{h_r} r_{t,i} \left[ U_i \mathbf{e}_t \right]. \tag{5}$$

We call this layer the role interaction layer (RIL). The role assignment vector $\mathbf{r}_t = \left[ r_{t,1}, \ldots, r_{t,h_r} \right]^\top$ indicates the role(s) $i \in [1, \ldots, h_r]$ to which the token $x_t$ is assigned. Each role corresponds to a subset of aspects encoded in the token embedding, and is extracted by linearly transforming the token embedding, i.e., $U_i \mathbf{e}_t$. These extracted role-specific token vectors are then weighted-summed using the role assignment vectors in Eq. (5). See Fig. 1 for a graphical illustration of the RIL.

## 3.1 ROLE ASSIGNMENT

The proposed RIL leaves the design of the role assignment computation in Eq. (4) open. In this paper we consider three different approaches to role assignment.

**Dense assignment**   We affine-transform the output $\hat{r}_t$ from the LSTM layer at each time step in Eq. (4) and apply a hyperbolic tangent element-wise:

$$\mathbf{r}_t^{\text{dense}} = \tanh(W\hat{r}_t + b).$$

We call this a dense role assignment, as it corresponds to assigning the token to all the roles (hence, dense) with varying degrees of assignment.

**Additive assignment: Softmax assignment**   Instead, we can constrain the role assignment to be additive with varying strengths. We use softmax normalization (Bridle, 1990):

$$r_{t,i}^{\text{softmax}} = \frac{\exp(\mathbf{u}_i^\top \mathbf{r}_t^{\text{dense}})}{\sum_{j=1}^{h_r} \exp(\mathbf{u}_j^\top \mathbf{r}_t^{\text{dense}})} > 0.$$

Unlike the dense assignment, the additive assignment, or softmax assignment, only allows "positive" assignment of a token to each role.

**Discrete assignment: One-hot assignment**   The softmax assignment above can be thought of as computing the categorical distribution over the $h_r$ roles. In other words, $r_{t,i}^{\text{softmax}}$ is the probability of the token being assigned to the role $i$:

$$p(r_{t,i}^{\text{onehot}} = 1 | X) = r_{t,i}^{\text{softmax}}. \tag{6}$$

---

[2] We omit the superscript, $X$ or $Y$, whenever it is unnecessary.

From this distribution, we can sample a single role during training, or choose the most likely role during test, to which the corresponding token is assigned. One difficulty arising from this discrete assignment is that we cannot use backpropagation due to the non-differentiability. In the experiments later, we use Gumbel-Softmax relaxation (Maddison et al., 2017; Jang et al., 2017) to address this issue during training.

**Role Assignment in the Target Side**     On the source side, the role assignment vector for each token can be computed using the entire context, that is, considering both the previous and future tokens. It is, however, not possible to do so on the target side, due to the autoregressive nature of conditional language modelling in Eq. (3). The role assignment of target tokens must be done only based on the previous tokens. To accommodate this restriction on the target side and the lack thereof on the source side, we use the unidirection LSTM on the target side and the bidirectional LSTM on the source side:

$$\mathbf{r}_t^X = f_{\text{role}}^X(\text{Bi-LSTM}(\mathbf{e}_1^X, \mathbf{e}_2^X, \ldots, \mathbf{e}_T^X)_t) \qquad \mathbf{r}_t^Y = f_{\text{role}}^Y(\text{LSTM}(\mathbf{e}_1^Y, \mathbf{e}_2^Y, \ldots, \mathbf{e}_{t-1}^Y)_t).$$

## 3.2 ROLE COMBINATION

We further consider two variants of the combination of role-specific token representations, $U_i\mathbf{e}_t$, from Eq. (5). These variants are introduced in order to address the high computational cost as well as to facilitate learning.

**Identity Role: Residual RIL**     In the original formulation, each role has its own fully-trainable transformation matrix $U_i$. This effectively increases the depth of the entire network by one and adds multiplicative interaction, both of which are known to make learning more difficult (Glorot & Bengio, 2010; Sutskever et al., 2011). We avoid this issue by introducing a fixed role with an identity transformation applied to every token. That is, $r_{h_r+1}^T = 1$, and $U_{h_r+1} = I$. This is equivalent to introducing a residual connection (He et al., 2016) that bypasses the proposed residual RIL:

$$\tilde{\mathbf{e}}_t = \sum_{i=1}^{h_r} r_{t,i}\left[U_i\mathbf{e}_t\right] + 1 \cdot \left[I\mathbf{e}_t\right] = \sum_{i=1}^{h_r} r_{t,i}\left[U_i\mathbf{e}_t\right] + \mathbf{e}_t. \tag{7}$$

**Rank-1 Tensor Approximation**     Although the number of roles $h_r$ is small, we introduce $h_r \times h \times h$ new parameters, which can be prohibitive or more easily leads to overfitting. We address this by approximating the third-order tensor $U = [U_1|U_2|\cdots|U_{h_r}] \in \mathbb{R}^{h_r \times h \times h}$ with three matrices, $U_r \in \mathbb{R}^{h_r \times h}$, $U_e\mathbb{R}^{h \times h}$ and $U_o \in \mathbb{R}^{h \times h}$. This leads to

$$\tilde{\mathbf{e}}_t = U_o\left[(U_r\mathbf{r}_t) \odot (U_e\mathbf{e}_t)\right] + \mathbf{e}_t, \tag{8}$$

where $\odot$ is an element-wise multiplication. We present it with the residual layer to show that the identity role is not considered a part of the third-order tensor in the rank-1 approximation.

## 4 EXPERIMENTAL SETUP

We test the proposed role interaction layer (RIL) on the problem of machine translation. As it is reasonable to assume that existing state-of-the-art neural translation systems are able to capture and disambiguate various roles/meanings of tokens implicitly, our goals are two-fold; first, to identify under which settings the explicit handling of contextual roles helps, and second, to understand what kinds of roles the proposed layer captures. In order to satisfy both of these goals, we evaluate the proposed approach on two language pairs–En-Fi and En-De– with two corpora of varying sizes– IWSLT (De-En) and WMT (En-De, En-Fi), while using state-of-the-art translation systems based on the recently proposed transformer (Vaswani et al., 2017).

**Data**     In the case of De-En, we use IWSLT'14, which contains approximately 160k sentence pairs, as a low-resource language pair (Cettolo et al., 2014). To test the effect of the proposed layer when there are abundant resources, we also test En-De on WMT'14 which contains approximately 4.5M sentence pairs. To contrast the effect of the proposed layer between different languages, we choose WMT'17 En-Fi which contains approximately 2.3M sentence pairs. Following (Ott et al., 2018), we tokenize and segment all sentences using byte pair encoding (BPE, Sennrich et al., 2016). We use vocabularies of 10k and 40k subwords for IWSLT and WMT, respectively.

**Baseline** We use the transformer (Vaswani et al., 2017) as a baseline system. In particular, we train a small model for IWSLT'14 De-En, a medium model for WMT'17 En-Fi and a big model for WMT'14 En-De. The small model has three layers on each of the encoder and decoder, and each transformer block has four attention heads and is 256 (512 for feedforward) dimensional. The medium model has six layers for both encoder and decoder with eight attention heads and is of 512 (2048) dimensional. The large model has 16 attention heads and is 1024 (4096) dimensional. We refer readers to (Vaswani et al., 2017) for more details.

The proposed RIL introduces contextualization of token representations (Peters et al., 2018) based on a small LSTM layer. In order to verify the contribution of this contextualization without multiplicative role interaction, we build a **contextualized** baseline which replaces the proposed RIL with $\tilde{\mathbf{e}}_t = U\mathbf{r}_t + \mathbf{e}_t$, where $U \in \mathbb{R}^{h \times h_r}$.

**Matched Baseline** The proposed RIL introduces additional parameters, which makes it difficult to gauge the contribution of the RIL from the increased model capacity. Hence, we also train a **matched** version of each of the aforementioned baseline systems by increasing the dimensionality to (approximately) match the number of parameters of our model with the RIL.

**Proposed Models** Our proposal is generic in that we can incorporate the proposed **RIL** layer into any existing neural network operating on a discrete sequence by simply inserting it between the embedding layer and the main part of a neural network. This is precisely the strategy we take here. For each baseline model, we insert the proposed layer and compare it against the baseline and its matched variant. We test each of the three role assignment strategies (**dense**, **softmax** and **one-hot**) from Sec. 3.1. We evaluate the following variants of the proposed RIL:

- **+Residual**: the fixed role with the identity transformation in Eq. (7)
- **+Approx**: rank-1 tensor approximation in Eq. (8)
- **+CNN**: replace the LSTM with CNN in the RIL

**Training** We closely follow the training strategy of Ott et al. (2018).[3] We use Adam (Kingma & Ba, 2015) with a linearly growing learning rate for the first 4,000 updates up to $10^{-3}$ for IWSLT'14 and $5 \times 10^{-4}$ for both WMT'14 and WMT'17. The learning rate is then decayed by the inerse square root of the number of updates. We use minibatches of up to 4k, 20k and 400k tokens for IWSLT'14, WMT'17 and WMT'14, respectively. We smooth the label with $\epsilon = 0.1$ (LeCun et al., 2012) and use dropout (Srivastava et al., 2014) of $0.3$ after each transformer block as well as on the token embedding and the role assignment vector across all the models. For each setting, we report the mean and standard deviation of the BLEU scores obtained from five training runs for IWSLT'14 and WMT'17, and three runs for WMT'14. We use beam search with beam width set to 4 for inference.

**Discrete Role Assignment** For the discrete assignment we initially train the model with the softmax assignment. We then finetune it using the Gumbel-softmax relaxation (Maddison et al., 2017; Jang et al., 2017). We decay the softmax temperature starting from 5 by 0.9995 after each update down to the minimum of $0.5$. We use a constant learning rate of $2 \times 10^{-4}$ for IWSLT and $10^{-4}$ for both WMT'14 and WMT'17 during fine-tuning.

## 5 RESULTS AND ANALYSIS

### 5.1 IWSLT'14 DE-EN

We conduct the most extensive set of experiments on the low-resource setting with WMT'14 De→En. Results are presented in Table 1. We make a number of interesting and important observations. First, our baseline (Transformer (Small)) and matched baseline (Transformer (Small, Matched)) are as good as or better than the existing state of the art reported by Edunov et al. (2017) and Elbayad et al. (2018). This gives us a strong confidence in reporting our experiments and results. Although we observe small improvement from contextualization (+contextualization), this improvement is as pronounced as that from simply increasing the number of parameters (matched).

---

[3] We use FairSeq (Gehring et al., 2017) for implementation and experimentation.

| Role Assignment | dense | | softmax | | onehot | |
|---|---|---|---|---|---|---|
| # of Roles $h_r$ | 16 | 32 | 16 | 32 | 16 | 32 |
| Transformer (Small) | 33.62 ±0.09 | | | | | |
| + contextualized | 34.08±0.14 | 34.31±0.15 | | | | |
| Tranformer (Matched) | 34.29±0.05 | | | | | |
| +RIL+Approx | 34.38±0.17 | 34.38±0.15 | - | - | - | - |
| +RIL | 34.54±0.16 | 34.67±0.06 | 34.59±0.06 | 34.61±0.08 | 34.67±0.10 | 34.56±0.07 |
| +RIL+Residual | 34.65±0.14 | 34.68±0.08 | 34.69±0.03 | **34.74±0.12** | 34.70±0.19 | 34.67±0.12 |
| +RIL+CNN+Residual | 34.44±0.14 | 34.60±0.12 | 34.58±0.07 | 34.66±0.11 | 34.57±0.07 | 34.51±0.12 |
| Edunov et al. (2017) | 32.23±0.10 | | | | | |
| Elbayad et al. (2018) | 33.81±0.03 | | | | | |

Table 1: Test results on IWSLT'14 De-En. We observe that the proposed RIL with the fixed role (+Residual) improves the translation quality up to 1 BLEU point over the baseline and 0.5 BLEU over the matched baseline. Both our baselines as well as the model with the proposed RIL clearly beat the previous state-of-the-art results obtained using comparable setups.

| | WMT'17 En-Fi | WMT'14 En-De |
|---|---|---|
| Transformer | 21.82±0.19 | 29.82±0.07 |
| Transformer (Matched) | 21.69±0.05 | 29.88 ±0.12 |
| +RIL (dense) | 21.89±0.17 | 29.78±0.08 |
| +RIL (softmax) | **22.08±0.12** | 29.47±0.18 |
| +RIL (dense) +Residual | 21.90±0.10 | **29.92±0.13** |
| +RIL (softmax) +Residual | 21.87±0.09 | 29.71±0.04 |
| Vaswani et al. (2017) | - | 28.0 |
| Dehghani et al. (2018) | - | 28.9 |
| Ott et al. (2018) | - | 29.3 |

Table 2: Test results on WMT'17 En-Fi and WMT'14 En-De. Based on the previous experiments on IWSLT'14 De-En, we use 32 roles and LSTM for the proposed RIL. The RIL improves the translation quality by a small margin over our own baselines which are up to 1.8 BLEU higher than the reported baselines from the literature in the case of En-De.

Most importantly, we observe that the proposed RIL improves upon these strong baselines. The improvement was most significant when the residual connection, via a fixed role with an identity transformation, was used in conjunction. The improvement was observed across all three types of role assignments with slight advantages when using the softmax (additive) assignment with 32 roles. The models, where the role encoding LSTM was replaced with the convolutional network (+CNN) of width 7, always underperformed their counterparts with the LSTM for the role assignment. This suggests that roles are best assigned when the entire context was taken into account.

## 5.2 WMT'14 EN-DE AND WMT'17 EN-FI

Based on the observations for IWSLT'14, we test dense and softmax role assignment strategies with 32 roles. As the CNN and rank-1 tensor approximation were not effective, we do not evaluate them in these larger-resource settings. We also do not consider the contextualized baseline. We present the results in Table 2. First, we again observe that our own baseline is stronger than any reported results on WMT'14 En-De. The improvement is up to 1.8 BLEU points.[4] This assures us of the fairness of the comparison. In this larger-resource regime, we observe that the improvement from the proposed RIL is substantially smaller than the previous experiment. We generally observe as before it is a good choice to use the residual variant of the RIL, and there is almost no difference between the dense and softmax role assignment strategies when the residual connection is used.

The diminished improvement may be explained by two possibilities. One possibility is that the RIL does not work as designed and does not have any effective contribution to modelling. The other is that the large transformer, when trained with a large amount data, already captures the various roles of tokens and does not benefit from explicit role interaction. In the latter case, the RIL may still capture various roles, as designed, and it may still give us a window through which we can take a peak at the internal mechanism behind the transformer, which we explore in the next section.

| R | Freq | Interpretation | Examples |
|---|---|---|---|
| 1 | 1251 | adjective modifier or verb prefix | 'have a negative impact', 'if it prohibits' |
| 3 | 1219 | noun at end of sentence | 'above sea level .', 'photo ID card .' |
| 5 | 301 | (ad)verb or conjunction at sentence/clause start | 'Over the past', 'care before it is' |
| 6 | 1263 | infix of proper noun | 'the Jedlicka', 'at Quantico' |
| 7 | 648 | noun suffix (typically of proper noun) | 'The Korean is', 'said Reyes' |
| 13 | 10087 | stop word | 'the', 'is', 'in', 'to', 'a' |
| 14 | 1108 | suffix | 'telescope', 'boredom', 'palliative' |
| 16 | 915 | prefix | 'telescope', 'palliative' |
| 19 | 608 | infix | 'constellation', 'irritated' |
| 22 | 4385 | highly overloaded: verb, prep. object, etc. | 'I think this', 'of military weapons ' |
| 28 | 2736 | punctuation, conjunction and start of sentence | 'It is a', '"It was', 'military weapons .' |
| 29 | 1399 | similar to role 3 and 22 | |
| 30 | 2023 | word, typically noun, preceded by 'the' | 'at the same time', 'of the time' |

Table 3: Source-side (English) roles discovered by the RIL trained on WMT'14 En-De. The discovered roles generally correspond to meaningful, context-dependent of tokens. Frequency is computed over the development set. We omit roles that are rarely used or hard to interpret.

| Roles | Function | Examples | Translation |
|---|---|---|---|
| 0, 23 | pronoun in inner clause | , vor der wir uns
, der neben ihr | , in front of which we
, the one next to her |
| 1 | article indicating possession (genitive) | ende der welt
Gefühl der Verwirrung | end of the world
feeling of confusion |
| 5, 9, 28 | article for indirect object (dative) | in der Wüste
zu der Erkenntnis kommen | in the desert
reaching the insight |
| 6 | pronoun | Teufel der spricht
der Gedanke der zählt | devil who talks
the thought that counts |
| 8,11 | article in nominative | der Präsident muss
der Markt ist | the president must
the market is |
| 30 | suffix as participle indicator | nachlassender Sehfähigkeit
treibender kultur | decreasing eyesight
driving culture |

Table 4: The case study of the German BPE token 'der' using the IWSLT'14 model. Based on the context, different roles are assigned to the single token 'der'.

### 5.3 Qualitative Analysis: Role Discovery

**Individual Roles: WMT'14 En-De**   We finetune RIL (softmax) + Residual models to train discrete role assignments and run the resulting system over the development sets from WMT'14 En-De to extract roles assigned to the source-side tokens. We manually inspect these role assignments to interpret each of the 32 roles. In Table 3, we list most of the 32 roles for which we could find suitable interpretation and which were assigned frequently enough. Although the context dependency makes it difficult to understand each role on its own, we nevertheless make some interesting observations.

First, many of the roles are used to capture multi-token structures, such as phrase-level structures or prefix-infix-suffix structures. For instance, the roles 14, 19 and 14 respectively correspond to the prefix, infix and suffix of long words that are often broken into multiple subwords by the BPE segmentation. Larger structures are also present: we observe many roles indicating the beginning and end of sentences or clauses, such as roles 3 and 28. Second, many of these roles could be interpreted as reflecting the tokens' syntactic roles. For instance, role 1 is active for adjective modifiers and role 5 for adverbs or conjunctions at the start of a clause. Interestingly, due to BPE encoding we find that entire words are categorized the same way as subwords. For example, the noun 'States' when preceded by 'United' is typically assigned role 7 which is typically used for noun suffixes. For the interested reader we provide a similar analysis for WMT'17 En-Fi in Table 5 of the appendix.

**Individual Tokens: IWSLT'14 De-En**   In order to investigate the effect of context on the role assignment, we turn to the model trained on IWSLT'14 De-En and inspect the German token 'der', which we know to have multiple translations based on its contextual function. In Table 4, we list

---

[4]The improvement over Ott et al. (2018) is largely due to the updated preprocessing routine in FairSeq.

some of the assigned roles and their functions. Inspecting these roles together with how 'der' was translated into English, the importance of the latent role discovered and captured by the RIL is apparent. It facilitates disambiguating various usages of 'der' by considering the context which is particularly important for a proper English translation. Again, these roles are highly syntactic.

## 6    RELATED WORK

**Variable Binding**    The role combination in Eq. (5) can be rewritten as

$$\tilde{\mathbf{e}}_t = \sum_{i=1}^{h_r} r_{t,i} \left[ U_i \mathbf{e}_t \right] = (\mathbf{r}_t \otimes \mathbf{e}_t) U,$$

where $\mathbf{r}_t \otimes \mathbf{e}_t \in \mathbb{R}^{h_r \cdots h}$ is a vectorized tensor product, and $U = [U_1; U_2; \cdots ; U_{h_r}] \in \mathbb{R}^{(h_r \cdot h) \times h}$ is the concatenation of our role-specific transformation matrices. This use of tensor product between the token representation and the role assignment reminds us of earlier work addressing the variable binding problem via tensor products (Smolensky, 1990; Clark & Pulman, 2007; Huang et al., 2018).

**Modulated Transformation in Neural Networks**    The (latent) role assignment in the RIL can be thought of as modulating linear transformations of the input token. This idea of using hidden variables to modulate linear transformation of one variable to another has a long tradition in deep learning. Memisevic & Hinton (2010) used such modulated connection to capture image transformation between a pair of input and output images, which was followed by Taylor & Hinton (2009) who applied this idea to temporal modelling of motion, where the latent variables captured motion style. This approach was used by Mnih & Hinton (2007) and Sutskever et al. (2011) for language modelling, both of which are the closest works to ours in this paper. The former only focused on the rank-1 approximation, and the latter assigned each input token to a separate role without having an mechanism to cluster multiple tokens based on their context into a small number of groups, as done in this paper. This idea was more recently applied to modelling compositionality of language (Socher et al., 2013b) and relational reasoning (Socher et al., 2013a).

**The Quality of Baselines**    One noticeable observation from our experiments is that our baseline outperforms the previously published results by some margin. The matched baseline outperforms the latest state-of-the-art translation quality achieved by the 2-D convolutional network on IWSLT'14 De-En, outperforms all the state-of-the-art results using transformers on WMT'14 En-De, and outperforms the best entry from WMT'17 on WMT'17 En-Fi (Östling et al., 2017).[5] This is due to our careful tuning of the baselines, and agrees well with recent observations on how poorly baselines are often treated (Melis et al., 2018; Merity et al., 2018).[6] Even with these well-tuned, strong baselines, the proposed RILs were found to generally further improve translation quality.

## 7    DISCUSSION AND CONCLUSION

In this paper, we proposed to explicitly incorporate context-dependent latent role interaction as a way to enrich the conventional embedding layer for token representation. We introduced the role interaction layer (RIL) which consists of two stages; role assignment and role combination. We evaluated the effect of the proposed RIL on machine translation using the state-of-the-art transformer-based system on two language pairs and two parallel corpora.

We make three notes about the proposed RIL when used for enriching the token representation in neural machine translation. First, we observed that this explicit way of incorporating role interaction through dedicated linear transformations improved the translation quality when the amount of data was limited, as in IWSLT'14 De-En and WMT'17 En-Fi. This implies that such role binding cannot be easily captured from a limited amount of data, but that we can encourage the model to do so by explicitly giving it the capability of role interaction via the proposed RIL. Second, a large state-of-the-art neural translation system, such as the one used in our experiments, does not benefit from the

---

[5] `http://matrix.statmt.org/matrix/output/1871?score_id=26781`
[6] See also `https://goo.gl/oqU1kJ` which illustrates a strong correlation between the reported baseline and claimed improvement in each published paper using the same dataset from the same conference.

proposed RIL when trained with a large amount of data, as in WMT'14 En-De. This was somewhat expected due to the capability of a main part of the network in capturing long-term dependencies across the entire context. This gives strong evidence that role interaction already happens implicitly in existing state-of-the-art translation systems. Even in this case, the proposed RIL is useful due to its inductive bias and resulting transparency. We can explicitly probe roles to which tokens on both source and target sides were assigned, and easily group them according to the role assignment. Lastly, we point out that the proposed RIL is not restricted to be used after the table lookup layer. Its design was carefully done in order to allow it to be inserted in any part of an existing neural network.

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

| R | Freq | Interpretation | Examples |
|---|---|---|---|
| 0 | 844 | adjective modifier or compound | 'at the same time', 'in the Middle East' |
| 2 | 1411 | noun at end of sentence or clause | 'the Middle East.', 'ten minutes, before' |
| 3 | 921 | infix of proper noun | 'said Wenger', 'from Kotka to' |
| 4 | 271 | infix of verb | 'are leveling off', 'are blasted with' |
| 5 | 839 | verb | 'Mugabe read out', 'be thanked for' |
| 9 | 2762 | (proper) noun prefixes or compound | 'Hewlett Packard', 'Robert Mugabe' |
| 10 | 1689 | word/enumeration after preposition | 'between Lamb and Schmidt' |
| 14 | 3108 | beginning (sometimes end) of sentence | 'There is no', 'Di Maria' |
| 15 | 980 | suffix | 'the 911 call', 'real estate' |
| 16 | 972 | words at beginning of clauses | 'but we will', ', which would' |
| 18 | 614 | prefix | 'Moody', 'the BBC' |
| 21 | 1774 | suffix | 'Square are', '''Wenger' |
| 24 | 7812 | conjunctions, punctuations and prepositions | 'to', 'and', 'Lamb's mind' |
| 25 | 1994 | articles | 'a', 'the' |
| 29 | 1229 | common nouns | 'your name', 'in a statement' |
| 30 | 524 | words after preposition, conjunction or punct. | 'food and grocery', 'in social media' |
| 31 | 6365 | stop word | 'the', 'be', 'not', 'just' |

Table 5: Source-side (English) roles discovered by the RIL trained on WMT'17 En-Fi. We observe that the discovered roles generally correspond to syntactic roles of tokens and that they are context-dependent. Frequency is computed over the development set. Roles that are rarely used or hard to interpret are omitted.

