# OpenReview forum: "Contextualized Role Interaction for Neural Machine Translation"
_ICLR.cc/2019/Conference_

### Official Review · AnonReviewer3 · 2018-11-01
**The improvement seems not enough**

**Rating:** 4
**Confidence:** 4

**Review:**

The paper proposes contextual role representation which is an interesting point.
The writing is clear and the idea is original.
Even with the interesting point, however, the performance improvement seems not enough compared to the baseline. The baseline might be carefully tuned as the authors said, but the proposed representation is supposed to improve the performance on top of the baseline.
The interpretation of the role representation is pros of the proposed model. However, it is somehow arguable, since it is subjective.

- minor issues:
There are typos in the notations right before Eq. (8).

---

> ### Author Response · Authors · 2018-11-08
> **Rebuttal**
>
> We thank you for your review.
>
> We believe that creating and analysing network variations for the sole purpose of improving some rather artificial scores on some benchmarks is limiting the kind of research we can conduct. By designing and carefully evaluating meaningful model extensions we can learn a lot about both the model extensions and the baseline. For instance, in our case we find that the RIL exhibits its intended behavior and improves the translation quality when there is a limited amount of data (which is an important use case for machine translation), although there is no visible improvement on larger datasets. We thus draw an interesting conclusion from this observation that the baseline can/must learn something similar implicitly *if* enough data is available.

---

### Official Review · AnonReviewer2 · 2018-11-02
**Interesting idea, but the improvement over the baseline is not significant.**

**Rating:** 5
**Confidence:** 4

**Review:**


[Summary]
This paper proposes “a role interaction layer” (briefly, RIL) that consists of context-dependent (latent) role assignments and role-specific transformations: Given an RIL layer, different dimensions of an embedding vector are “interacted” based on Eqn. (5), Eqn. (7), etc. The authors work on IWSLT De->En and WMT En->De, En->Fi to verify their proposed algorithm with case study included.

[Pros]
(+) I think the idea/thought of using a “role interaction layer” is interesting.  The case study in Section 5.3 demonstrates different “roles”. Also, different RIL architectures are designed.
(+) The paper is easy to follow.

[Cons & Details]
(1) As stated in the abstract, “…, but that the improvement diminishes as the size of data grows, indicating that powerful neural MT systems are capable of implicitly modeling role-word interaction by themselves…” (1) The main concern is that, considering RIL does not obtain significant gain on large datasets, then we cannot say that the proposed algorithm is better than the baseline. (2) Why the NMT systems trained on large dataset can “implicitly modeling role-word interaction”, while small dataset cannot? Any analysis?

(2) For the “matched baseline”, page 5, you increase the dimensionality of the models. But an RIL is an additional layer, which makes the network deeper. Therefore, a baseline with an additional layer should be implemented.

---

> ### Author Response · Authors · 2018-11-08
> **Rebuttal**
>
> We thank you for your thorough review.
>
> “we cannot say that the proposed algorithm is better than the baseline.”
>
> We agree that we cannot say that our addition is better than the baseline on the larger datasets we considered.
>
> “Why the NMT systems trained on large dataset can “implicitly modeling role-word interaction”, while small dataset cannot?”
>
> We conjecture that a neural translation system does not exhibit an appropriate inductive bias that encourages it to capture role interaction and cannot learn to do so when only a small amount of data is available. The proposed approach equips the neural translation model with the appropriate inductive bias that allows it to exploit role interaction to do better. However, with a large enough data, our observation suggests that this explicit inductive bias is unnecessary as the neural translation system can learn this role interaction property on its own from data.

---

> > ### Comment · AnonReviewer2 · 2018-11-22
> > **Reply to the rebuttal**
> >
> > Thanks for your response.
> >
> > 1.	The improvement is not significant on large datasets. It seems that you agree with this point in your rebuttal. AnonReviewer1 also points it out in his/her ``[significant] comments’’. Therefore, this is still a concern for this paper. You should try to make it work before the revision deadline.
> >
> > 2.	I did not find the detailed analysis of “implicitly modeling role-word interaction”. It seems that no revised paper is uploaded until now. I am still confused about your explanation: (a) What is the “inductive bias” ? (b) You mentioned that “we conjecture that a neural translation system does not exhibit an appropriate inductive bias that encourages it to capture role interaction and cannot learn to do so when only a small amount of data is available.” Any statistics/visualization to support this claim? From my point of view, this explain is weak.
> >
> >
> > 3.	You have not implemented the baseline with an additional layer, i.e., the second point in my “[Cons & Details]” comments. Any results ?

---

### Official Review · AnonReviewer1 · 2018-11-02
**The motivation and goal of this paper are unclear**

**Rating:** 4
**Confidence:** 5

**Review:**



[Summary]
This paper proposes a “role interaction layer” (RIL) to capture the context-dependent (latent) role for each token.


[clarity]
The writing is basically clear.
However, It is hard for me to get the motivation and goal of this paper.
Is the main purpose of the proposed method “improving the performance” or “interpretability”?


[originality]
It seems that the proposed method consists of several known methods.
Moreover, even though the purpose differs, technically the proposed method is closely related to the method proposed in [Shu+,2018].
Therefore, the technical novelty of the proposed method is limited.

[Shu+,2018] Raphael Shu, Hideki Nakayama, “Compressing Word Embeddings via Deep Compositional Code Learning”, ICLR-2018.


[significance]
The contribution of this paper is not very clear.
The improvements from the baseline method (Matched) is less than 1 point BLEU as shown in Table 1 and 2, which is not a significant improvement.



Overall, this paper is basically well written. However, this paper seems a technical report rather than a conference paper.

---

> ### Author Response · Authors · 2018-11-08
> **Rebuttal**
>
> We thank you for your thorough review.
>
> “Is the main purpose of the proposed method “improving the performance” or “interpretability”?”
> “The contribution of this paper is not very clear.”
> “The improvements from baseline method (Matched) is less than 1 point BLEU as shown in Table 1 and 2, which is not a significant improvement.”
>
> The purpose of this work was to give existing architectures a better inductive bias. The results of such an effort can be either or both better performance or/and better interpretability.  A stronger inductive bias typically leads to better generalization, which is often more necessary in the low-data regime, and our experiments clearly show that the proposed approach indeed improves the translation quality (better performance) when the amount of data is limited. Even when it does not offer better performance due to the availability of large data, the proposed RIL facilitates the interpretation of a model, which provides us with a deeper understanding of what these neural translation models are learning.
>
> “technically the proposed method is closely related to the method proposed in [Shu+,2018].”
>
> The method proposed by Shu et al. [2018] is indeed related to our proposal. Our role assignments are however learned end-to-end with the downstream task while Shu et al. learn the codes of pre-trained embedding matrices. Furthermore, using discrete codes is just one possible formulation of the proposed approach in addition to other possibilities we explore in our work.

---

> > ### Comment · AnonReviewer1 · 2018-11-26
> > **Response to the rebuttal**
> >
> > Thank you for the response.
> >
> > According to the authours' response,  now I understand that the proposed method is mainly for the "low-data regime."
> > However, we cannot find this kind of descriptions in the submitted version.
> > I believe that this paper should be re-organized to clarify the primal goal (or motivation) of the proposed method as I wrote in the first review.
> >
> > There is no additional information that I should consider to improve the overall recommendation score.
> > Therefore, I decided to keep my score unchanged.

---

### Meta-Review · Area_Chair1 · 2018-12-13
**Some clarity issues, and improvements underwhelming**

**Confidence:** 4
**Recommendation:** Reject

**Metareview:**

This paper proposes to improve MT with a specialized encoder component that models roles. It shows some improvements in low-resource scenarios.

Overall, reviewers felt there were two issues with the paper: clarity of description of the contribution, and also the fact that the method itself was not seeing large empirical gains. On top of this, the method adds some additional complexity on top of the original model.

Given that no reviewer was strongly in favor of the paper, I am not going to recommend acceptance at this time.